# Brd4-Brd2 isoform switching coordinates pluripotent exit and Smad2-dependent lineage specification

Rosalia Fernandez-Alonso[1], Lindsay Davidson[2], Jens Hukelmann[3], Michael Zengerle[4], Alan R Prescott[3], Angus Lamond[3] (iD), Alessio Ciulli[4], Gopal P Sapkota[1] & Greg M Findlay[1,*] (iD)

## Abstract

Pluripotent stem cells (PSCs) hold great clinical potential, as they possess the capacity to differentiate into fully specialised tissues such as pancreas, liver, neurons and cardiac muscle. However, the molecular mechanisms that coordinate pluripotent exit with lineage specification remain poorly understood. To address this question, we perform a small molecule screen to systematically identify novel regulators of the Smad2 signalling network, a key determinant of PSC fate. We reveal an essential function for BET family bromodomain proteins in Smad2 activation, distinct from the role of Brd4 in pluripotency maintenance. Mechanistically, BET proteins specifically engage Nodal gene regulatory elements (NREs) to promote Nodal signalling and Smad2 developmental responses. In pluripotent cells, Brd2-Brd4 occupy NREs, but only Brd4 is required for pluripotency gene expression. Brd4 downregulation facilitates pluripotent exit and drives enhanced Brd2 NRE occupancy, thereby unveiling a specific function for Brd2 in differentiative Nodal-Smad2 signalling. Therefore, distinct BET functionalities and Brd4-Brd2 isoform switching at NREs coordinate pluripotent exit with lineage specification.

**Keywords** BET Bromodomain; differentiation; embryonic stem cell; Nodal-Smad2 signalling; Pluripotency

**Subject Categories** Signal Transduction; Stem Cells

## Introduction

Embryonic/pluripotent stem cells (ESCs/PSCs) have the developmental capacity to differentiate into any cell type within the adult body, including terminally differentiated tissues such as pancreas, liver, neurons and cardiac muscle [1]. Thus, directed differentiation is of significant clinical interest due to potential applications in regenerative therapeutics. Although many insights into the signalling pathways controlling differentiation have been gleaned from developmental genetics and embryology [2], the molecular events which coordinate pluripotent exit with developmental progression are poorly understood.

Transforming growth factor (TGF)-β signalling plays a prominent role both in maintaining pluripotency and in mesendoderm (ME) specification [3–5], which is a developmentally and therapeutically significant population [2]. The TGFβ superfamily consists of TGFβ isoforms and Activin/Nodal ligands, which selectively engage TGFβII receptors. This leads to TGFβI receptor activation, Smad2/3 recruitment and phosphorylation, and Smad2/3-dependent gene expression in complex with Smad4 [3–5]. In PSCs, autocrine Nodal activates Smad2 signalling [6] to maintain expression of pluripotency factors [7,8]. In differentiating cells, Nodal-Smad2 induces ME-specific genes [9,10] and reinforces ME fate selection by suppressing neuroectoderm (NE) genes [11,12] (Fig 1A). However, in contrast to our detailed understanding of the core Nodal-Smad2 pathway and transcriptional targets, we remain poorly informed about the molecular mechanisms by which Smad2 signalling is established and maintained in diverse developmental contexts.

Here, we use a library of potent and selective small molecule inhibitors to uncover the regulatory network surrounding the Nodal-Smad2 pathway in differentiating PSCs. This approach identifies bromodomain and extra-terminal (BET) proteins as unanticipated regulators of Nodal-Smad2 signalling, distinct from the role of Brd4 in pluripotency maintenance. We find that BET family members Brd2 and Brd4 directly engage Nodal gene regulatory elements (NREs) to drive autocrine Nodal expression. Using quantitative proteomics to explore BET family expression dynamics during differentiation, we show that Brd4 suppression facilitates pluripotent exit and drives increased Brd2 recruitment to NREs. Brd2 thereby selectively governs Smad2 pathway activation in differentiating cells and is required for ME differentiation. Our data uncover a

1   The MRC Protein Phosphorylation and Ubiquitylation Unit, School of Life Sciences, The University of Dundee, Dundee, UK
2   Pluripotent Stem Cell Facility, School of Life Sciences, The University of Dundee, Dundee, UK
3   Centre for Gene Regulation and Expression, School of Life Sciences, The University of Dundee, Dundee, UK
4   Biological Chemistry and Drug Discovery, School of Life Sciences, The University of Dundee, Dundee, UK
    *Corresponding author. Tel: +44 1382 386066; E-mail: g.m.findlay@dundee.ac.uk

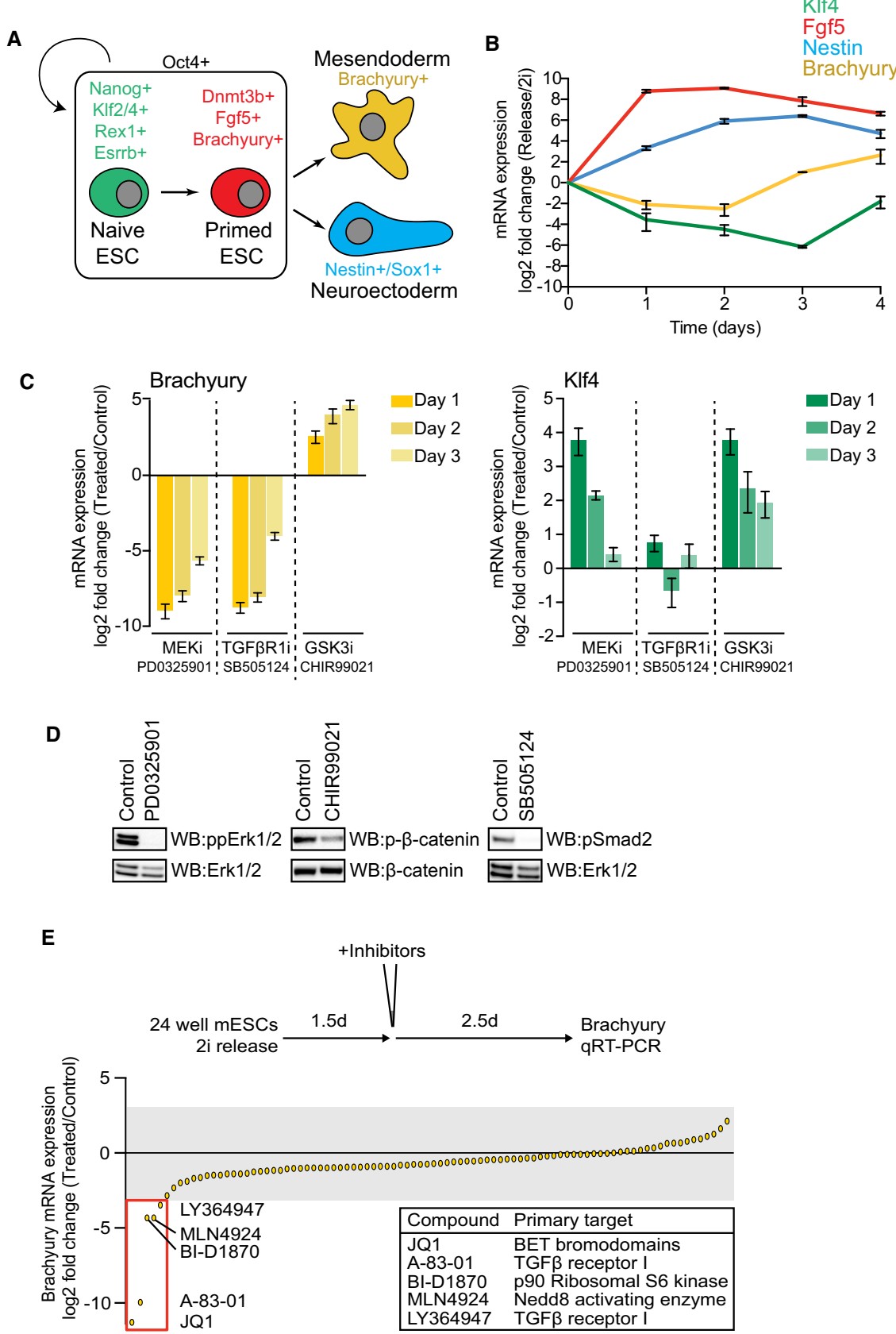

**Figure 1.**

**Figure 1. A chemical screen to identify novel players in the Smad2 pathway.**

A   Schematic representation of mESC differentiation towards mesendoderm (ME) and neuroectoderm (NE) fates. Nodal-Smad2 signalling controls ME specification.

B   Klf4, Fgf5, Nestin and Brachyury mRNA levels in mESCs differentiating upon 2i release were quantified by qRT–PCR and normalised to mRNA levels in 2i mESCs. Data are presented as mean $\pm$ SEM of technical replicates from a representative experiment; similar results were observed in three independent experiments ($n = 3$).

C   mESCs differentiating upon 2i release for 4 days were treated with PD0325901 (MEK1/2 inhibitor; 1 μM), CHIR99021 (GSK3 inhibitor; 3 μM) and SB505124 (TGFβRI inhibitor; 3 μM) on either day 1, 2 or 3 as indicated. mRNA levels of Klf4 and Brachyury were quantified by qRT–PCR and normalised to DMSO control. Data are presented as mean $\pm$ SEM of technical replicates from three independent experiments ($n = 3$).

D   mESCs differentiating upon 2i release for 1.5 days were treated with 1 μM PD0325901, 3 μM CHIR99021 or 3 μM SB505124 for a further 2.5 days. Smad2 Ser465/467, β-catenin Ser33/37/41 and Erk1/2 Thr202/Tyr204 phosphorylation and total Erk1/2 or β-catenin were determined by immunoblotting ($n = 3$).

E   Screen of Brachyury mRNA levels following mESC differentiation with inhibitors (1 μM; JQ1 100 nM) added for the final 2.5 days. Data are presented as mean of technical replicates. Small molecules and primary targets which robustly inhibit Brachyury induction are highlighted.

Source data are available online for this figure.

## Results

### A screen for modifiers of Nodal-Smad2 pathway activity during mESC differentiation

In order to elucidate the extended regulatory network surrounding the core Nodal-Smad2 pathway in differentiating PSCs, we employed mouse ESCs (mESCs) maintained in Mek1/2 (PD0325901) and Gsk3 (CHIR99021) inhibitors, referred to as "2i" [13]. Upon 2i release [14], naïve ESCs (Klf4) undergo differentiation towards epiblast (Fgf5), followed by lineage specification of NE (Nestin) and ME (Brachyury; Fig 1B [15]). ME differentiation requires Smad2 pathway activation [16,17], as Brachyury induction is suppressed by treatment with the TGFβI receptor inhibitor SB505124 [18] (Fig 1C). Brachyury is also regulated by Fgf-Mek1/2 and Wnt signalling in the expected manner, as the Mek1/2 inhibitor PD0325901 suppresses, whilst the Gsk3 inhibitor CHIR99021 promotes Brachyury expression (Fig 1C). Each inhibitor appropriately suppresses signalling pathway output (Fig 1D).

We therefore exploited Brachyury induction as a biological assay for Nodal-Smad2 signalling in differentiating mESCs. A screen of 105 potent and selective tool compounds identifies several small molecules which potently suppress Brachyury induction (Fig 1E, see also Appendix Table S1). In this cohort are TGFβI receptor inhibitors A-83-01 and LY364947, the Nedd8 activating enzyme inhibitor MLN4924, the p90 Ribosomal S6 Kinase (RSK) inhibitor BI-D1870 and the bromodomain and extra-terminal (BET) inhibitor JQ1.

### BET bromodomain activity is required for Nodal-Smad2 signalling in differentiating mESCs

Initially, we investigated whether hits identified in our screen directly affect Nodal-Smad2 signalling. MLN4924 suppresses expression of multiple lineage markers (Fig EV1A), whilst a structurally distinct p90 RSK inhibitor SL0101 does not inhibit Brachyury expression (Fig EV1B), suggesting that MLN4924 and BI-D1870 have non-specific effects in differentiating mESCs. However, JQ1 specifically abolishes Smad2 phosphorylation without inhibiting Fgf-Mek1/2 or Wnt signalling (Fig 2A), demonstrating that the Nodal-Smad2 pathway is a functional target of BET activity in differentiating mESCs. We corroborate this observation using other BET inhibitors including IBET and the structurally unrelated IBET-151 and PFI-1 (Fig 2B) [19,20]. In contrast, RVX-208, a specific inhibitor of the second BET bromodomain [21], does not significantly suppress Smad2 phosphorylation (Fig 2B).

We asked whether diverse BET inhibitors consistently block Brachyury induction in differentiating mESC. Suppression of Brachyury by BET inhibitors closely correlates with inhibition of Smad2 phosphorylation (Fig 2C). We also confirm that JQ1 suppresses expression of Nanog, a known target gene (Fig EV1C) and Brachyury protein (Fig EV1D). Furthermore, JQ1 inhibits direct Smad2 transcriptional targets and ME markers Mixl and Gsc (Fig EV1E) and strongly promotes Sox1$^+$ NE lineage selection in a manner similar to TGFβI receptor inhibition (Fig 2D) [22]. In contrast, JQ1 does not suppress Smad1 phosphorylation, a specific readout of BMP signalling (Fig EV1F), indicating that BET inhibition specifically blocks the TGFβ-Smad2 axis of the TGFβ/BMP signalling network. Taken together, our data provide substantial evidence that BET inhibitors block Nodal-Smad2 dependent signalling during mESC differentiation.

### BET function patterns autocrine Nodal expression to underpin Smad2 pathway activation

Initially, we speculated that BET inhibitors suppress Nodal-Smad2 signalling by inhibiting a kinase. However, JQ1 does not inhibit > 140 kinases, including the TGFβI receptor Alk5 [23,24]. Consistent with this finding, BET inhibition suppresses Smad2 phosphorylation only after 48 h (Fig EV2A), suggesting that BET proteins control Nodal-Smad2 signalling via a transcriptional mechanism. In order to identify BET-dependent transcripts in the TGFβ and Wnt signalling networks, we interrogated published RNAseq data from PSCs (Gene Expression Omnibus GSE60171) [22]. This analysis indicates that NODAL and INHBE ligands and the Smad2-dependent negative feedback regulator LEFTY2 are repressed following BET inhibition (Fig 3A), suggesting a potential mechanism by which BET proteins function in Nodal-Smad2 signalling.

To test this hypothesis, we conducted a targeted mRNA expression analysis of TGFβ signalling components in mESCs following 2i release (Fig 3B). As expected, Smad2 transcriptional targets Brachyury, Mixl and Lefty1/2 are potently suppressed by JQ1 [25]. Strikingly, JQ1 treatment also suppresses expression of Activin/Inhibin and Nodal type TGFβ ligands, whilst TGFβ family

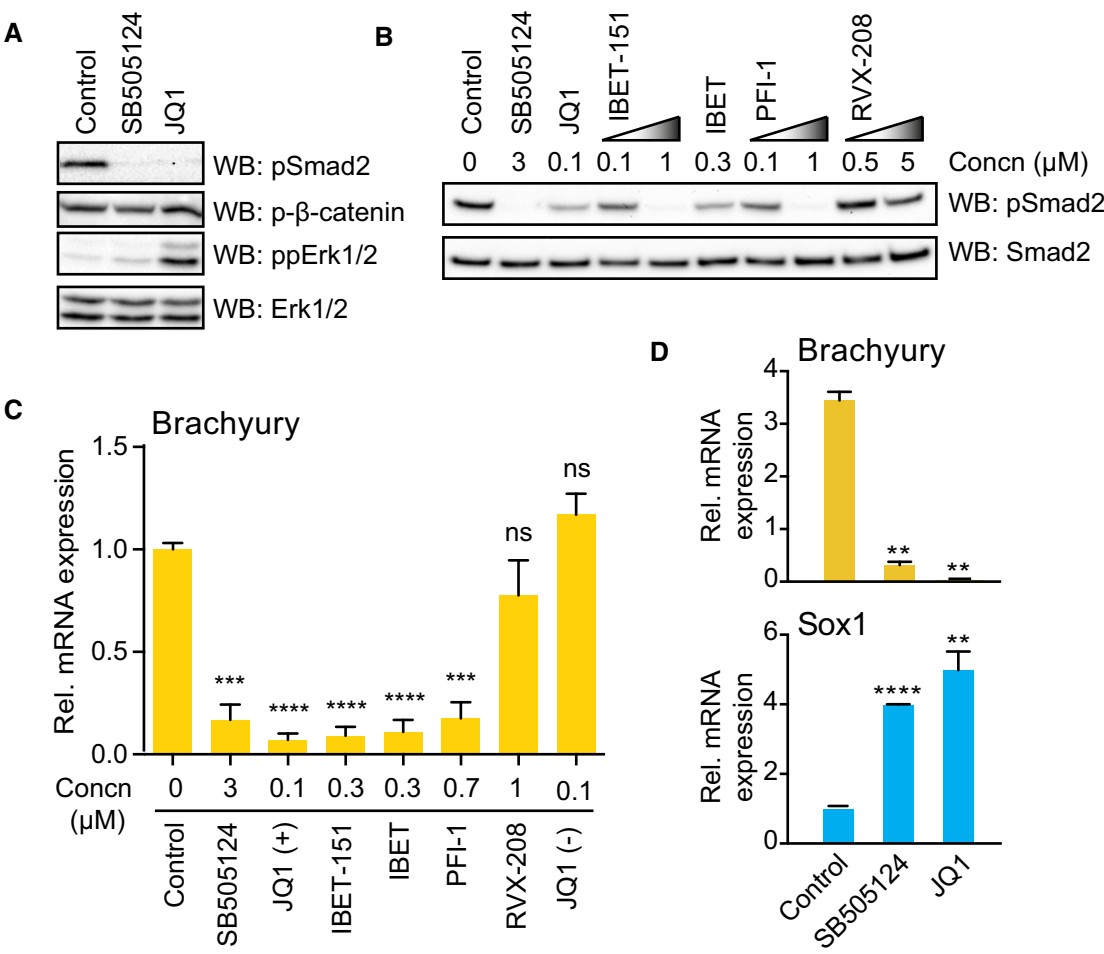

**Figure 2. BET bromodomain activity is required for Smad2 signalling and mESC differentiation.**

A mESCs differentiating upon 2i release for 1.5 days were treated with 3 μM SB505124 or 100 nM JQ1 for a further 2.5 days. Smad2 Ser465/467, β-catenin Ser33/37/41 and Erk1/2 Thr202/Tyr204 phosphorylation and total Erk1/2 were determined by immunoblotting (n = 3).

B mESCs differentiating upon 2i release for 2 days were treated with the indicated inhibitor concentrations. Smad2 phosphorylation and total Smad2 were evaluated by immunoblotting (n = 3).

C mESCs differentiating upon 2i release for 1.5 days were treated with the indicated inhibitors for a further 2.5 days. Brachyury mRNA levels were determined by qRT–PCR and normalised to DMSO control. Data are presented as mean ± SEM of technical replicates from three independent experiments (n = 3). Two-tailed unpaired Student's *t*-test; ns = not significant, ***P < 0.001 and ****P < 0.0001.

D mESCs differentiating upon 2i release for 1.5 days were treated with 3 μM SB505124 or 100 nM JQ1 for a further 2.5 days. Brachyury and Sox1 mRNA levels were determined by qRT–PCR. Data presented as mean ± SEM of technical replicates from three experiments (n = 3). Two-tailed unpaired Student's *t*-test (**P < 0.01 and ****P < 0.0001).

Source data are available online for this figure.

receptor/co-receptor expression is only modestly altered or increased by JQ1 (Fig 3B). Selective suppression of Activin/Nodal ligands may therefore underpin Nodal-Smad2 inhibition by JQ1. Nodal is prominently expressed during early development [26] and in differentiating mESCs (Fig 3C), suggesting that BET proteins drive Nodal expression and resulting Smad2 signalling in differentiating mESCs.

In order to determine whether Activin/Nodal ligand expression is the key BET-dependent step in the Smad2 pathway, we tested whether recombinant Activin/Nodal ligand can rescue Smad2 signalling in JQ1 treated mESCs. We were unable to source or produce active recombinant Nodal [27]; however, Activin A uses the same TGFβ receptor family as Nodal [28,29]. Accordingly, Activin A

treatment robustly restores Smad2 phosphorylation in JQ1-treated cells, in contrast to TGFβ1 or BMP4 (Fig 3D). Activin A treatment also rescues expression of ME markers Brachyury (Fig 3E) and Mixl (Fig EV2B) following JQ1 treatment, whilst TGFβ1 and BMP4 do not. Expression of Nodal itself is not rescued by Activin A treatment (Fig EV2B), highlighting a specific defect in autocrine Nodal production. These data also confirm that JQ1 does not inhibit the Trim33 bromodomain containing protein, which is required downstream of Nodal-Smad2 for ME differentiation [27]. We therefore propose that BET inhibitors block Smad2 signalling and PSC differentiation by suppressing Activin/Nodal production. This mechanism is specific to Nodal signalling, as JQ1 does not inhibit TGFβ1-Smad2 dependent PAI-1 transcription in U2OS cells [30] (Fig EV2C).

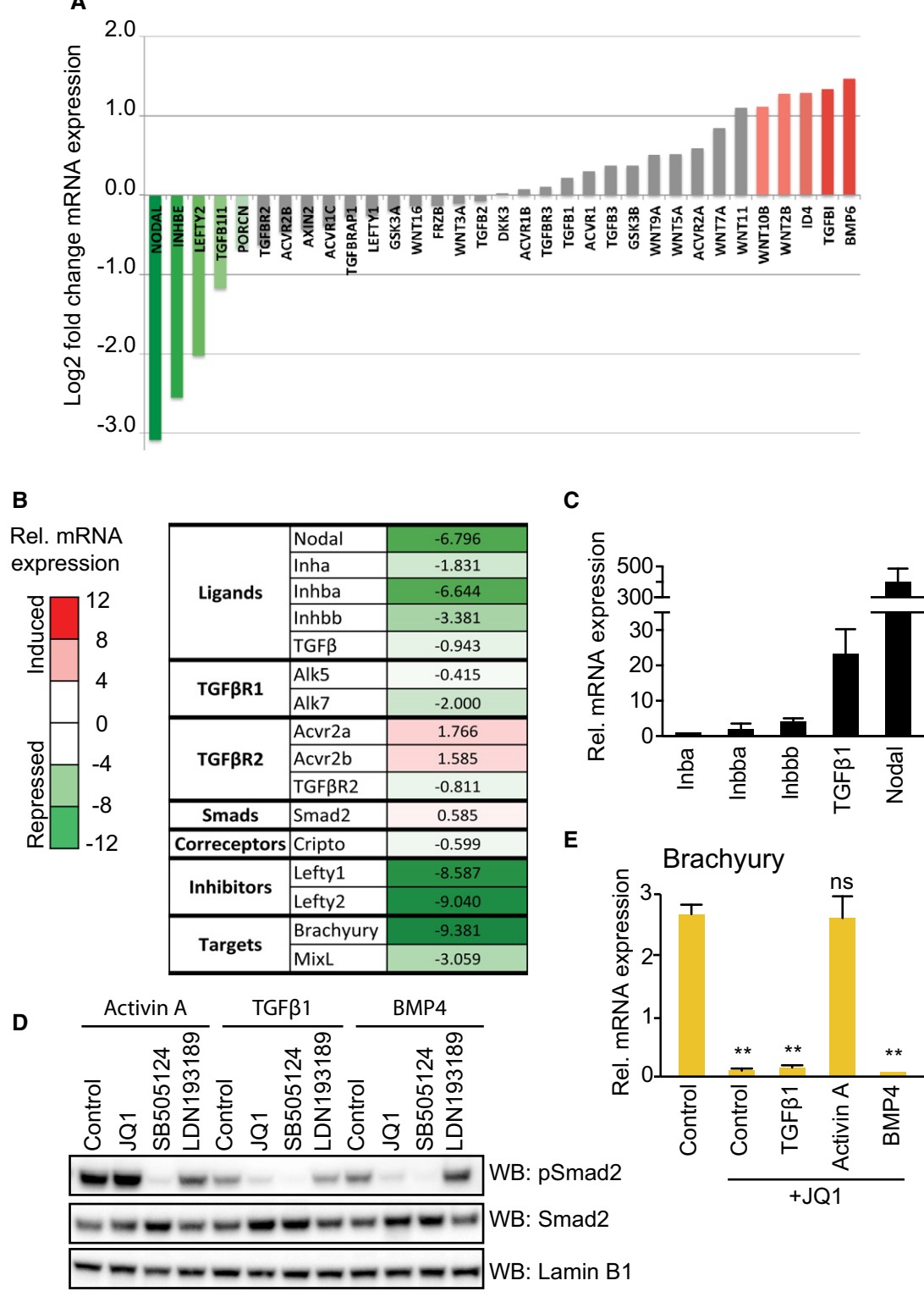

**Figure 3.**

## BET-Nodal-Smad2 signalling modulates directed differentiation of hiPSCs to dEN

Next, we explored the wider developmental and therapeutic significance of Nodal-Smad2 regulation by BET family bromodomains.

Specification of human-induced pluripotent stem cells (hiPSCs) to definitive endoderm (dEN) is a key intermediate step in pancreatic and hepatic development [2]. Differentiation of Sox17[+]/FoxA2[+] dEN from hiPSCs is initiated by the Wnt activator CHIR99021 followed by Activin A treatment (Fig 4A). In this system, CHIR99021 drives Smad2

**Figure 3.   BET bromodomain function patterns autocrine Nodal expression and signalling.**

A   Graphical representation of RNAseq data (GSE60171) corresponding to a list of TGFβ-Smad2 and Wnt pathway genes. Data represent the log2 fold change in mRNA expression of hESCs treated with MS436 BET inhibitor compared to control. Genes repressed by BET inhibition are shown in green and induced genes are in red.

B   mESCs differentiating upon 2i release for 1.5 days were treated with 100 nM JQ1 for a further 2.5 days, and levels of the indicated transcripts determined by qRT–PCR. Table shows the mean fold change in mRNA expression, with genes repressed by JQ1 in green and the induced ones in red. Data from a representative experiment are shown. Similar results were observed in three independent experiments (*n* = 3).

C   Relative mRNA expression of the indicated TGFβ superfamily ligands determined by qRT–PCR. Data are presented as mean ± SD of technical replicates. Similar results were observed in three independent experiments (*n* = 3).

D   mESCs differentiating upon 2i release were treated with 100 nM JQ1, 3 µM SB505124 or 1 µM LDN193189 for 2.5 days and then stimulated with Activin A, TGFβ1 or BMP4 for 30 min. Smad2 phosphorylation, total Smad2 and Lamin B1 levels were determined by immunoblotting. Data from a representative experiment are shown. Similar results were observed in three independent experiments (*n* = 3).

E   mESCs differentiating upon 2i release were treated with 100 nM JQ1 for 2.5 days and stimulated with TGFβ1, Activin A and BMP4 for 48 h. Brachyury mRNA levels determined by qRT–PCR. Data are presented as mean ± SD of technical replicates. Statistical significance was determined for each condition relative to control using two-tailed unpaired Student's *t*-test; ns = not significant, **P* < 0.01. Similar results were found in three independent experiments (*n* = 3).

Source data are available online for this figure.

phosphorylation and dEN differentiation [31], and this is abolished by JQ1 treatment (Fig EV3A). Furthermore, CHIR99021 induces Nodal expression within 6 h, which is similarly blocked by JQ1 (Fig 4B). BET proteins are therefore required for autocrine Nodal-Smad2 signalling not only in mESCs, but also in differentiating hiPSCs.

Having established the importance of BET proteins for Nodal-Smad2 signalling in hiPSCs, we examined whether BET activity is required for specification of hiPSCs to dEN. hiPSC treatment with SB505124 or JQ1 suppresses Brachyury induction (Fig EV3B), indicating a failure to specify ME. As a result, treatment of hiPSCs with JQ1, even for the first 24 h of differentiation, completely abolishes dEN specification, as assessed by dEN-specific markers FoxA2 and Sox17 (Fig 4C) and loss of dEN morphology (Fig 4D). In contrast, JQ1 does not significantly alter morphology of undifferentiated hiPSC colonies (Fig 4D). Loss of dEN identity in JQ1-treated samples is confirmed by immunostaining for FoxA2 (Fig 4E). Our data therefore show that BET proteins govern activation of Nodal-Smad2 signalling in hiPSCs, which plays a critical role in differentiation of therapeutically relevant dEN progenitor cells.

**Nodal gene regulatory elements (NREs) form a putative BET protein recruitment platform**

We then sought to elucidate the mechanism by which BET proteins drive autocrine Nodal-Smad2 signalling. As key transcriptional regulators, we hypothesised that BET proteins are recruited to Nodal gene regulatory elements (NREs) to control Nodal expression. The Nodal promoter consists of multiple NREs, including the key highly bound element (HBE) 0.5–2.5 kb upstream [32] and the asymmetric

enhancer (ASE) 1.0 kb downstream of the transcriptional start site (Fig EV4A). We therefore examined whether NREs contain lysine-acetylated histone H4 BET bromodomain binding motifs [24], which could in principle directly recruit BET family members [33]. Indeed, chromatin immunoprecipitation (ChIP) using an anti-acetyl K5/K8/K12/K16 histone H4 antibody enriches for the Nodal HBE (Fig EV4B), confirming that the Nodal promoter is a potential recruitment site for BET family bromodomain proteins.

**Determining BET family expression dynamics by quantitative proteomics**

In order to determine which BET family members might be recruited to NREs in differentiating mESCs, we conduct quantitative whole cell proteomics to determine the relative expression of BET family members. We quantify total protein copy number per cell from mESCs in LIF/FCS. This analysis identified all members of the BET family: Brd2, Brd3, Brd4, and Brdt. Brd2 (204,033 ± 30,770 copies per cell) and Brd4 (142,198 ± 1,436 copies per cell) are the major isoforms expressed in mESCs in LIF/FCS (Fig 5A), although Brd3 is also expressed at significant levels (92,437 ± 2,483 copies per cell). The testis-specific isoform Brdt is expressed at very low levels in mESCs (135 ± 17.8 copies per cell).

**Brd2 and Brd4 engage NREs to modulate Nodal gene transcription**

Our finding that Brd2 and Brd4 are abundantly expressed in mESCs (Fig 5A) prompted us to determine whether Brd2 and

**Figure 4.   BET-Nodal-Smad2 signalling axis modulates directed differentiation of hiPSCs to definitive endoderm.**

A   Experimental workflow for directed differentiation of Sox17+/FoxA2+ definitive endoderm (dEN) from human-induced pluripotent stem cells (hiPSCs).

B   hiPSCs treated with 3 µM CHIR99021 were co-treated with DMSO (Control), 3 µM SB505124 or 100 nM JQ1 for 24 h and RNA or protein was collected at the indicated time points. Nodal mRNA was quantified by qRT–PCR and represented as mean ± SD of technical replicates. Smad2 phosphorylation and total Smad2 protein levels were determined by immunoblotting and quantified using ImageLab software. Similar results were found in three independent experiments (*n* = 3).

C   hiPSCs undergoing dEN differentiation were treated with SB505124 or JQ1 for either the first 24 h ("pulse") or for the whole dEN differentiation process ("sustained"). Levels of FoxA2 and Sox17 mRNAs were determined by qRT–PCR. Data are presented as mean ± SD of technical replicates. Statistical significance was determined for each condition relative to control using two-tailed unpaired Student's *t*-test (**P* < 0.05, ***P* < 0.01, ****P* < 0.001 and *****P* < 0.0001). Similar results were found in three independent experiments (*n* = 3).

D   hiPSCs undergoing dEN differentiation were treated with SB505124 or JQ1 for either the first 24 h ("pulse") or for the whole dEN differentiation process ("sustained"). Phase contrast images were obtained at 10× objective. Similar results were found in three independent experiments (*n* = 3).

E   hiPSCs undergoing dEN differentiation were fixed and immunostained for FoxA2 and DAPI stained to identify nuclei. Images were taken using a confocal microscope at 10× objective. Right panel shows a zoomed in view of an area from the respective images on the left panel.

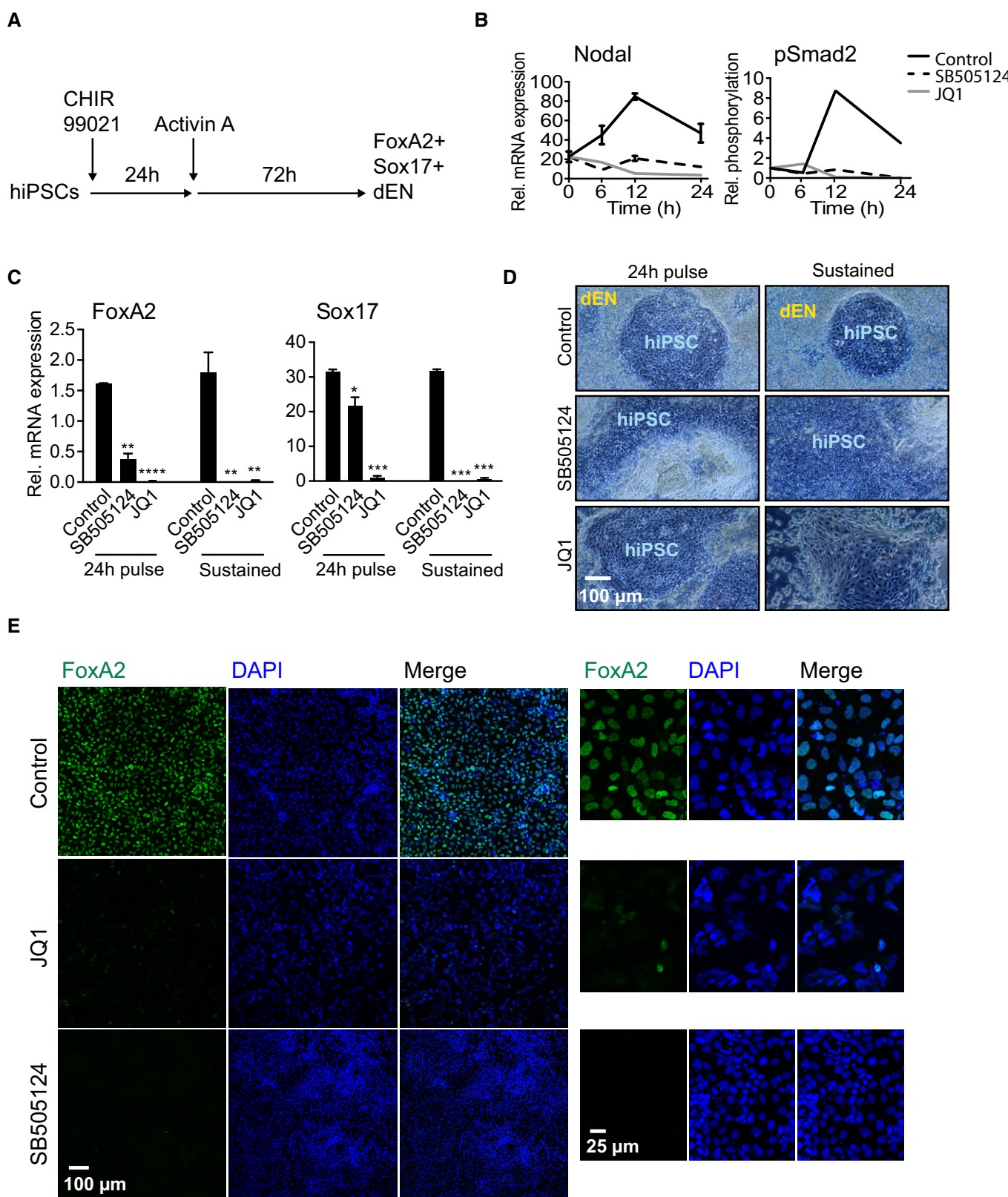

**Figure 4.**

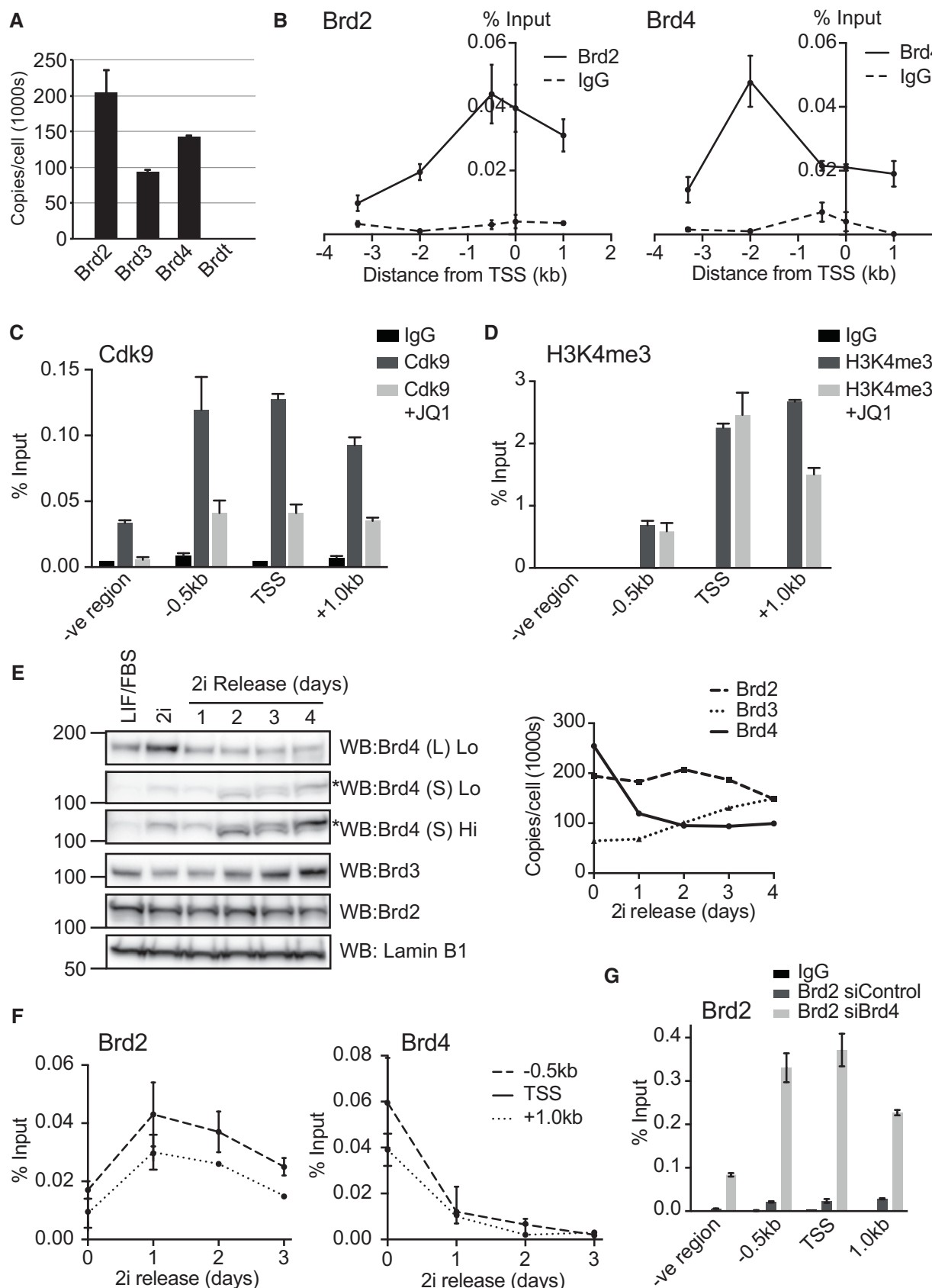

**Figure 5.**

**Figure 5.   BET protein expression dynamics during mESC differentiation underpin differential engagement to Nodal regulatory regions.**

A   Absolute abundance of BET family members (copy number/cell) was determined in mESCs cultured in LIF/FCS using quantitative whole cell proteomics. Data are presented as mean ± SD of technical replicates from three independent experiments (*n* = 3).

B   ChIP was performed on mESCs differentiating upon 2i release using anti-Brd2- and anti-Brd4-specific antibodies or respective IgG controls. Abundance of NRE genomic sequences was quantified by qPCR and the percentage of ChIP DNA compared to input calculated. Data are presented as mean ± SEM of technical replicates from a representative experiment. Similar results were obtained in three independent experiments (*n* = 3).

C   ChIP was performed on mESCs differentiating upon 2i release using anti-Cdk9 antibody or IgG control following treatment with 100 nM JQ1 or DMSO for 48 h. Abundance of NRE genomic sequences was quantified by qPCR and the percentage of ChIP DNA compared to input calculated. Data are presented as mean ± SEM of technical replicates from a representative experiment. Similar results were obtained in three independent experiments (*n* = 3).

D   ChIP was performed on mESCs differentiating upon 2i release using anti-histone H3K4me3 antibody or IgG control following treatment with 100 nM JQ1 or DMSO for 48 h. Abundance of NRE genomic sequences was quantified by qPCR and the percentage of ChIP DNA compared to input calculated. Data are presented as mean ± SEM of technical replicates from a representative experiment. Similar results were obtained in three independent experiments (*n* = 3).

E   Brd2, Brd3 and Brd4 protein levels were determined by immunoblotting in 2i mESCs or a time-course of mESCs differentiating upon 2i release. Low and high exposures of Brd4(S) are provided (Lo and Hi, respectively). A non-specific band in Brd4(S) is highlighted by an asterisk (left panel). ImageLab software was used for image acquisition and densitometric analysis, and values were compared to LIF/FCS control and proteomic data to extrapolate Brd2, Brd3 and Brd4 copy number/cell at each time point. Graph represents the number of copies of the indicated BET protein over time (Right panel). Similar results were obtained in three independent experiments (*n* = 3).

F   ChIP was performed on mESCs using Brd2 and Brd4 antibodies and the respective IgG controls every day during the course of 2i release. Abundance of NRE genomic sequences was quantified by qPCR, and the percentage of ChIP DNA over input was calculated. Data are presented as mean ± SEM of technical replicates from a representative experiment. Similar results were found in more than three independent experiments.

G   ChIP was performed on 2i mESCs using anti-Brd2 antibody or IgG control following transfection with control or Brd4 siRNA. Abundance of NRE genomic sequences was quantified by qPCR and the percentage of ChIP DNA compared to input calculated. Data are presented as mean ± SEM of technical replicates from a representative experiment. Similar results were obtained in three independent experiments (*n* = 3).

Source data are available online for this figure.

Brd4 directly engage NREs. Brd2- and Brd4-specific chromatin immunoprecipitation (ChIP) from differentiating mESCs enriches all NREs examined, including the HBE (–2.0 kb), transcriptional start site (0.0 kb) and ASE (1.0 kb), when compared to IgG control (Fig 5B). The NRE enrichment by Brd2 and Brd4 ChIP is highly selective, as it is abolished by MZ1-mediated degradation of BET proteins (Fig EV4C; note that NRE enrichment by Brd4 is significantly higher in 2i than in differentiating mESCs). MZ1 is a proteolysis targeting chimera (PROTAC) compound which chemically tethers BET proteins to the Von Hippel Lindau (VHL) E3 ligase, thereby eliciting specific degradation of Brd4 [34,35]. Importantly, Brd2 and Brd4 ChIP does not significantly enrich a distal genomic region reported as a low-level binder of BET bromodomain proteins [36] (Fig EV4D).

Our data suggest that BET proteins engage NREs to directly control Nodal gene transcription. Consistent with this notion, we show that JQ1 suppresses transcription of nascent Nodal mRNA, in addition to reducing steady-state levels (Fig EV4E). JQ1 also inhibits recruitment of Cdk9, which is required for transcriptional elongation,

to the Nodal transcriptional start site (Fig 5C), suggesting a molecular mechanism by which BET inhibition disrupts Nodal transcription. In contrast, JQ1 does not alter deposition of activating H3K4me3 marks at NREs, indicating that key epigenetic features of the Nodal gene remain unchanged upon BET inhibition (Fig 5D). Taken together, our data demonstrate that Brd2-4 directly control Nodal gene expression by regulating transcriptional machinery at NREs.

## BET family isoform switching alters Brd2-4 Nodal occupancy during mESC differentiation

Next, we sought to determine whether Brd2 and/or Brd4 are required for Nodal-Smad2 signalling during ME specification. To this end, we analysed expression dynamics of BET isoforms during mESC differentiation. Relative expression of BET family members in LIF/FCS mESCs and 2i mESCs at different stages of differentiation was evaluated by immunoblotting whole cell extracts (Fig 5E). We then extrapolated LIF/FCS mESC quantitative proteomics data to

**Figure 6.   Specific roles for Brd4 and Brd2 in pluripotency and mesendoderm differentiation of mESCs.**

A   2i mESCs were transfected with control, Brd2, Brd3 or Brd4 siRNAs, and phase contrast images were taken after 48 h at 10× objective. Similar results were obtained from three independent experiments (*n* = 3).

B   2i mESCs were transfected with control, Brd2, Brd3 or Brd4 siRNAs and Nanog, Brd2, Brd3, Brd4 and Lamin B1 protein levels evaluated by immunoblotting. Similar results were obtained from three independent experiments (*n* = 3).

C   2i mESCs were treated with the indicated concentrations of MZ1 for 4 h and Brd4, Brd2, Nanog and Lamin B1 protein levels determined by immunoblotting. A non-specific band in Brd4(S) is highlighted by an asterisk (left panel). Similar results were obtained from three independent experiments (*n* = 3).

D   mESCs differentiating upon 2i release were transfected with control, Brd2, Brd3 or Brd4 siRNAs and cultured for 72 h. Nodal and Brachyury mRNA levels were determined by qRT–PCR. Box plots show technical replicates from at least three independent experiments. Boxes extend from the 25th to the 75th percentile, with the median represented as a line segment inside each box. The whiskers expand from the minimum to the maximum value. Statistical significance was determined for each condition relative to control using two-tailed unpaired Student's *t*-test (ns = not significant, **P < 0.01 and ***P < 0.001).

E   ChIP was performed on mESCs differentiating upon 2i release using anti-Cdk9 antibody or IgG control following transfection with control or Brd2 siRNA. Abundance of NRE genomic sequences was quantified by qPCR and the percentage of ChIP DNA compared to input calculated. Data are presented as mean ± SEM of technical replicates from a representative experiment. Similar results were obtained in three independent experiments (*n* = 3).

F   ChIP was performed on mESCs differentiating upon 2i release using anti-histone H3K4me3 antibody or IgG control following 48 h transfection with control or Brd2 siRNA. Abundance of NRE genomic sequences was quantified by qPCR and the percentage of ChIP DNA compared to input calculated. Data are presented as mean ± SEM of technical replicates from a representative experiment. Similar results were obtained from three independent experiments (*n* = 3).

Source data are available online for this figure.

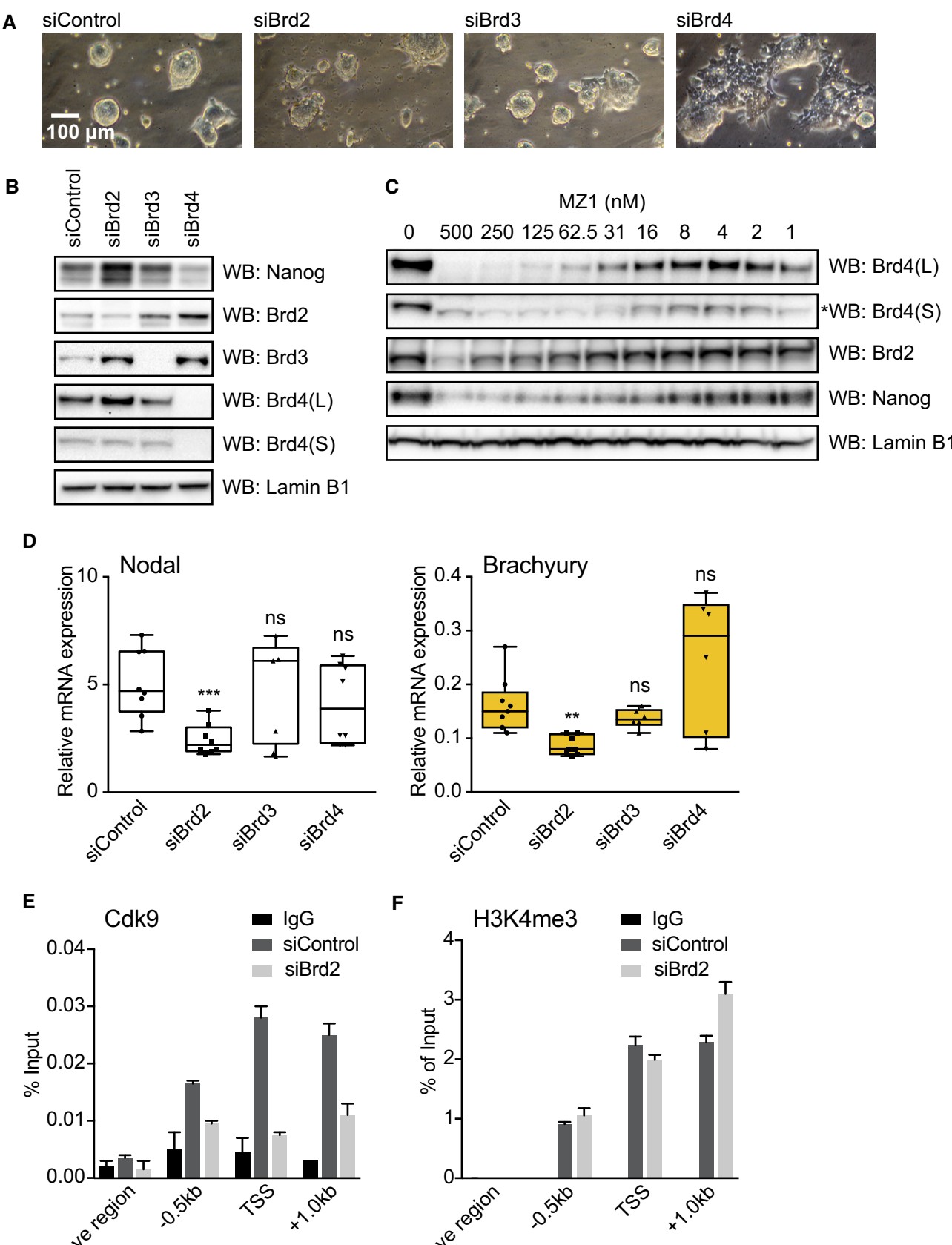

**Figure 6.**

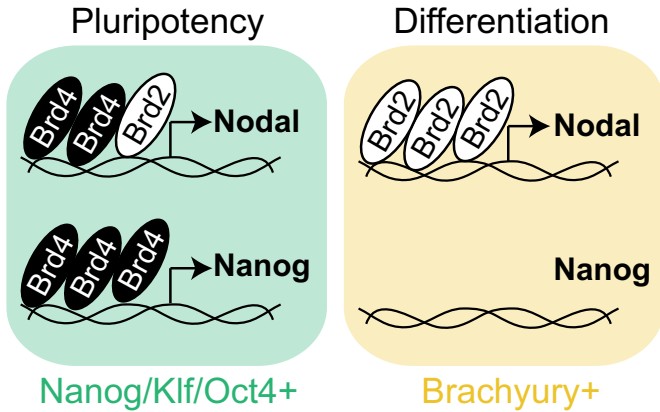

## Pluripotency

Nanog/Klf/Oct4+

## Differentiation

Brachyury+

**Figure 7.  Isoform switching and Brd2-4 functional dynamics coordinate mESC exit from pluripotency with Nodal-Smad2 dependent mesendoderm specification.**

determine the protein copy number per cell of Brd2, Brd3 and Brd4 short and long isoforms (Brd4 (S) and (L), respectively) in 2i and during release by relative quantification of immunoblotting data. This analysis indicates that Brd4 is the most abundant isoform in 2i mESCs (~250,000 protein copies per cell), followed by Brd2 (~200,000 copies per cell; Fig 5E). However, 2i release leads to rapid and sustained suppression of Brd4, stabilising at ~100,000 copies per cell. In contrast, Brd2 expression remains relatively constant at ~200,000 copies per cell (Fig 5E).

Our results suggest that BET family expression dynamics during mESC differentiation may reduce Brd4 bound NREs whilst increasing Brd2 occupancy. To test this possibility, we compared Brd2 and Brd4 engagement of NREs during a time-course of mESC differentiation following 2i release. Strikingly, NREs are highly enriched in Brd4 ChIP from 2i mESCs, and this is reduced in a graded fashion during mESC differentiation (Fig 5F). In contrast, NREs are modestly enriched in Brd2 ChIP from 2i ESCs, but Brd2 NRE occupancy increases upon differentiation (Fig 5F). Our data therefore indicate that Brd4-Brd2 "isoform switching" during mESC differentiation leads to loss of Brd4 and consequent enrichment of Brd2 at key Nodal gene regulatory elements. To directly test this, we examine Brd2 recruitment to NREs following Brd4 siRNA depletion. Strikingly, Brd4 knockdown augments Brd2 recruitment to NREs in 2i mESCs (Fig 5G), confirming that Brd2 and Brd4 compete for regulatory binding sites at the Nodal promoter.

### Brd2 specifically promotes autocrine Nodal expression and mesendoderm specification in differentiating mESCs

An isoform switch whereby Brd2 recruitment to NREs increases following Brd4 downregulation could, in principle, coordinate pluripotency exit with Nodal-Smad2 dependent mesendoderm differentiation. This hypothesis predicts that Brd4 suppression drives pluripotent exit [22], enabling Brd2 to become a specific and essential driver of Nodal-Smad2 signalling and ME differentiation. We addressed this by depleting individual BET family members by siRNA in 2i or differentiating mESCs. As predicted, knockdown of Brd2 or Brd3 does not significantly alter 2i mESC morphology (Fig 6A). However, Brd4 knockdown promotes a flattened

morphology characteristic of differentiating mESCs [22] (Fig 6A). Furthermore, Brd4 knockdown in 2i mESCs potently suppresses Nanog expression (Fig 6B) and drives induction of the neural marker Sox1 (Fig EV5A), confirming exit from pluripotency. In contrast, Brd2 or Brd3 knockdown have minor effects on Nanog (Fig 6B) and Sox1 expression (Fig EV5A), confirming the specific role of Brd4 in pluripotency maintenance [22]. We obtain similar results using MZ1 under conditions where Brd4 is specifically degraded (Fig 6C) [34], confirming that Nanog expression and pluripotency maintenance are Brd4 dependent.

Our finding that BET family isoform switching occurs at NREs during pluripotent exit implicates Brd2 as an essential driver of Nodal-Smad2 signalling and ME differentiation. Indeed, knockdown of Brd2, but not Brd3 or Brd4, significantly suppresses Nodal expression in differentiating mESCs (Fig 6D). Accordingly, specific knockdown of Brd2 significantly suppresses ME differentiation, as measured by expression of Brachyury (Fig 6D), Mixl, Lefty1 or Lefty2 (Fig EV5B). We quantify the level of siRNA knockdown achieved for Brd2-4 (Fig EV5C) and confirm that multiple Brd2 siRNA sequences significantly suppress Nodal and Brachyury expression (Fig EV5D). Our data therefore argue that Brd2 is a critical regulator of Nodal transcription and signalling in differentiating mESCs. Consistent with this notion, Brd2 is required for recruitment of Cdk9, a key driver of transcriptional elongation, to NREs (Fig 6E). However, Brd2 is not required for deposition of activating H3K4me3 marks at NREs, indicating that key epigenetic features of the Nodal gene remain unaltered (Fig 6F). In summary, our results propose a model whereby Brd4 depletion drives pluripotent exit and BET family isoform switching at the Nodal promoter, whereupon Brd2 comes to prominence as a critical regulator of autocrine Nodal-Smad2 signalling and ME specification (Fig 7).

## Discussion

Nodal-Smad2 is a central signalling pathway in ESC/PSC biology and plays a critical role during early development [26]. However, the mechanisms governing activation of the Nodal-Smad2 signalling network during development remain poorly understood. Here, we show that BET family bromodomain proteins are specifically required for Nodal signalling, Smad2 activation and ME differentiation of pluripotent cells. Our findings speak to a general concept that key transcriptional components such as BET bromodomains perform highly specific functions in different biological contexts. This notion is supported by recent work indicating that BET proteins frequently function at lineage-specific genes [37]. Interestingly, specific roles of BET family proteins appear to be conferred via the differential repertoires of co-factors expressed in cells with distinct developmental destinies.

In this study, we uncover a dynamic regulatory system, which is critical to coordinate pluripotent exit with early lineage specification. This is underpinned by "isoform switching" in the BET family, whereby Brd4 downregulation extinguishes pluripotency gene expression and initiates pluripotent exit [22,25]. We show that Brd2 then comes to prominence during differentiation, replacing Brd4 at Nodal promoter elements to drive autocrine Nodal-Smad2 signalling required for ME differentiation. An important facet of this switch is that Brd4 and Brd2 display specific functional capabilities. Only Brd4

supports Nanog expression, whereas Brd2 is able to drive Nodal expression and ME differentiation in the absence of Brd4. This mechanism may be underpinned by specific functions of Brd4 at distal enhancers [33] or super-enhancers, which are required for the expression of Nanog and other key developmental genes [38–40]. In contrast, core promoter elements of active genes are frequently co-occupied by Brd2, Brd3 and Brd4 together with RNAPolII and Med1 [33], which is consistent with Brd2 and Brd4 occupancy observed at proximal NREs. Importantly, we show that Brd2 binding to proximal NREs is required for Cdk9 recruitment and is sufficient for Nodal expression and Smad2 signalling during PSC differentiation.

It has been reported that BET inhibition induces ESCs to differentiate to neural lineages [22]. This may be explained by our finding that BET inhibition disables Smad2 signalling in PSCs. "Dual Smad" inhibition (referring to blockade of both TGFβ and BMP signalling pathways) is frequently used to efficiently differentiate neurons *in vitro* [41]. Furthermore, the BET bromodomain inhibitor IBET-151 forms part of a cocktail of small molecules used to directly convert fibroblasts to neurons [42]. Interestingly, IBET-151 functions by suppressing expression of fibroblast-specific genes, which may reflect loss of fibroblast identity upon Smad2 inhibition. Our data suggest that BET inhibition represents an effective strategy to disrupt Activin/Nodal signalling during neural differentiation, as this approach blocks ligand production and signal initiation, rather than inhibiting receptors or Smad2 directly, which may be rather less efficient and/or short-lived.

Finally, our results imply that BET bromodomains may control Smad2 signalling in disease contexts. Nodal overexpression is associated with epithelial to mesenchymal transition and invasiveness in certain cancers [43], suggesting that disease pathologies caused by deregulated Activin/Nodal might be reversed by BET inhibition. BET inhibitors could be readily exploited in this therapeutic setting, as they are already approved for treatment of cancers where oncogene expression (e.g. c-Myc) displays a strong dependency upon Brd4 [44].

# Materials and Methods

### Cell culture and differentiation

CGR8 mouse embryonic stem cells (ESCs) were maintained in 2i medium, DMEM/F12-Neurobasal (1:1), 0.5% N2, 1% B27 (Thermo-Fisher Scientific), 1% ʟ-glutamine, 100 μM β-mercaptoethanol, 3 μM CHIR99021 and 1 μM PD0325901. To induce differentiation, cells were plated at $4 \times 10^4$ cells/cm² in "2i release" medium (2i medium without CHIR99021 and PD0325901). Human-induced pluripotent stem cells (hiPSCs) were maintained in DEF-CS (Cellartis AB) and seeded on matrigel (10 μg/cm²) in DEF-CS for 3–4 days. To induce dEN differentiation, medium was changed to RPMI containing 25 mM HEPES and 3 μM CHIR99021 for 24 h, then replaced by RPMI-1640/Activin A (25 ng/ml) for 24 h prior to culture in RPMI/1x B27/Activin A (25 ng/ml) for 2 days.

### Small molecule screen

2i mESCs were plated at $8 \times 10^4$ cells in 24-well plates and differentiated in 2i release medium for 1.5 days. Media was then replaced by fresh media containing inhibitors or DMSO control and cells grown for a further 2.5 days prior to lysis, RNA extraction and quantitative RT–PCR (qPCR) analysis as described below.

### Immunoblotting

Cells were washed with phosphate-buffered saline (PBS) and lysed in ice-cold lysis buffer [20 mM Tris–HCl pH7.4, 150 mM NaCl, 1mM EDTA, 1% NP-40, 0.5% Na deoxycholate, 10 mM β-Glycerophosphate, 50 mM NaF, 10 mM NaPPi, 2 mM Na orthovanadate, 10% glycerol and complete protease inhibitor tablets (Roche)]. Protein extracts were quantified by BCA assay, and equal amounts of protein (10–30 μg) were subjected to SDS/PAGE, transferred to PVDF membranes and immunoblotted (see Appendix Table S2).

### Immunofluorescence and microscopy

hiPSCs were fixed in 4% (v/v) formaldehyde in PBS at room temperature for 10 min, washed twice with PBS, post-fixed with ice-cold 90% methanol for 5 min at –20°C, washed with PBS twice and blocked in 4% BSA in Tris-buffered saline with 0.1% Tween-20 (TBS-T) at room temperature for 1 h. Cells were incubated in primary antibody at 4°C overnight. After three washes with TBS-T, secondary antibody (Life Technologies) was added and incubated at room temperature for 1 h. After three washes with TBS-T, DAPI was added as a nuclear counterstain. Images were taken using a confocal microscope (Zeiss LSM 700).

### RNA extraction and qRT–PCR

RNA was extracted using the OMEGA total RNA kit and reverse transcribed using iScript reverse transcriptase (Bio-Rad) according to the manufacturer's instructions. qPCR was then performed using SsoFast™ EvaGreen® Supermix (Bio-Rad). See Appendix Table S3 for a list of primers. $\Delta C_t$ values using GAPDH as a reference gene were used to analyse the relative expression, and the $2^{-\Delta\Delta Ct}$ (Livak) method was used to normalise to control when required.

### Proteomic quantification of BET proteins

*Sample preparation and TMT labelling*
Cell pellets were lysed in 400 μl lysis buffer (4% SDS, 50 mM TEAB pH 8.5, 10 mM TCEP), boiled and sonicated with a BioRuptor (30 cycles: 30 s on, 30 s off) before alkylation with iodoacetamide for 1 h at room temperature in the dark. Lysates were subjected to the SP3 procedure for protein clean-up [45] before elution into digest buffer (0.1% SDS, 50 mM TEAB pH 8.5, 1 mM CaCl₂) and digest with LysC and Trypsin at a 1:50 (enzyme:protein) ratio. TMT labelling and peptide clean-up were performed according to the SP3 protocol. Samples were eluted into 2% DMSO, combined and dried in vacuo.

*Basic reverse-phase fractionation*
TMT samples were fractionated using offline high pH reverse-phase chromatography: samples were loaded onto a 4.6 × 250 mm Xbridge™ BEH130 C18 column with 3.5 μm particles (Waters). Using a Dionex BioRS system, the samples were separated using a 25-min multistep gradient of solvents A (10 mM formate at pH 9 in 2% acetonitrile) and B (10 mM ammonium formate pH 9 in 80%

acetonitrile), at a flow rate of 1 ml/min. Peptides were separated into 48 fractions, consolidated into 24, dried and peptides redissolved in 5% formic acid and analysed by LC-MS.

*Liquid chromatography electrospray tandem mass spectrometry analysis (LC-ES-MS/MS)*

1 μg of protein per fraction was analysed using an Orbitrap Fusion Tribrid mass spectrometer (Thermo Scientific) equipped with a Dionex ultra high-pressure liquid chromatography system (nano RSLC). RP-LC was performed using a Dionex RSLC nano HPLC (Thermo Scientific). Peptides were injected onto a 75 μm × 2 cm PepMap-C18 pre-column and resolved on a 75 μm × 50 cm RP-C18 EASY-Spray temperature controlled integrated column-emitter (Thermo) using a 4-h multistep gradient from 5% B to 35% B with a constant flow of 200 nl/min. The mobile phases were: 2% ACN incorporating 0.1% FA (Solvent A) and 80% ACN incorporating 0.1% FA (Solvent B). The spray was initiated by applying 2.5 kV to the EASY-Spray emitter, and the data were acquired under the control of Xcalibur software in a data-dependent mode using top speed and 4 s duration per cycle, the survey scan is acquired in the Orbitrap covering the *m/z* range from 400 to 1,400 Th with a mass resolution of 120,000 and an automatic gain control (AGC) target of 2.0 e5 ions. The most intense ions were selected for fragmentation using CID in the ion trap with 30% CID collision energy and an isolation window of 1.6 Th. The AGC target was set to 1.0 e4 with a maximum injection time of 70 ms and a dynamic exclusion of 80 s. During the MS3 analysis for more accurate TMT quantifications, 10 fragment ions were co-isolated using synchronous precursor selection using a window of 2 Th and further fragmented using HCD collision energy of 55%. The fragments were then analysed in the Orbitrap with a resolution of 60,000. The AGC target was set to 1.0 e5, and the maximum injection time was set to 300 ms.

*Database searching and reporter ion quantification*

The data were processed, searched and quantified with the MaxQuant software package, version 1.5.3.30. Proteins and peptides were identified using the UniProt *mouse* reference proteome database (SwissProt and Trembl accessed on 24.03.2016) and the contaminants database integrated in MaxQuant using the Andromeda search engine [46,47] with the following search parameters: carbamidomethylation of cysteine and TMT modification on peptide N-termini and lysine side chains were fixed modifications, whilst methionine oxidation, acetylation of N-termini of proteins, conversion of glutamine to pyro-glutamate and phosphorylation on STY were variable modifications. The false discovery rate was set to 1% for positive identification of proteins and peptides with the help of the reversed mouse Uniprot database in a decoy approach. Copy numbers of BET proteins were calculated as described [48] after allocating the summed MS1 intensities to the different experimental conditions according to their fractional MS3 reporter intensities.

## siRNA silencing

mESCs were transfected with 30 nM ON-TARGET plus SMART-pool or two individual sequences for either Brd2, Brd3 or Brd4, and with the SMART-pool non-targeting control siRNA (Dharmacon) using Lipofectamine RNAimax (Invitrogen) following the manufacturer's instructions. RNA or protein were extracted after 72 h for further analyses.

## Chromatin immunoprecipitation

Cells were cultured as described above and treated with 1 μM MZ1 [34] for 3 h where indicated. Crosslinking was performed by 1% (w/v) formaldehyde treatment for 10 min at room temperature and terminated by addition of 0.125 M glycine. Cells were washed with PBS twice and lysed in 50 mM Tris/HCl (pH 8.1), 10 mM EDTA, complete protease inhibitor cocktail (Roche) and 1% (w/v) SDS. Chromatin was sheared by fifteen 30-s pulses (30 s pause between pulses) at 4°C in a waterbath sonicator (Bioruptor, Diagenode). Soluble whole cell extract was diluted 10-fold in 20 mM Tris/HCl (pH 8.1), 2 mM EDTA, 150 mM NaCl and 1% (v/v) Triton X-100, and pre-cleared for 2 h at 4°C with 20 μl either protein A-agarose (rabbit antibodies) or protein G-agarose (sheep antibodies) beads and 2 μg of sheared salmon sperm DNA. Sample was incubated overnight at 4°C with 5 μg of the appropriate antibodies or 5 μg of control IgG, then for 1 h at 4°C on a rotating platform with 30 μl of either protein A-agarose (rabbit antibodies) or protein G-agarose (sheep antibodies). Beads were washed in wash buffer 1 (20 mM Tris/HCl (pH 8.0), 2 mM EDTA, 150 mM NaCl, 0.1% SDS and 1% (v/v) Triton X-100), wash buffer 2 (20 mM Tris/HCl (pH 8.0), 2 mM EDTA, 500 mM NaCl, 0.1% SDS and 1% (v/v) Triton X-100) then wash buffer 3 (10 mM Tris/HCl (pH 8.0), 1 mM EDTA, 0.25 M LiCl, 1% (v/v) Nonidet P40 and 1% (w/v) sodium deoxycholate), and twice with TE (10 mM Tris/HCl pH 8.0 and 1 mM EDTA). Bound complexes were eluted from the beads with 0.1 M NaHCO$_3$ and 1% (w/v) SDS, and crosslinking reversed by overnight incubation at 65°C in 0.2 M NaCl. Whole cell extracts were also subjected to crosslinking reversal. All samples were then incubated with RNase for 1 h and proteinase K (Roche) for 1 h at 45°C, and purified by Spin Column PCR purification kit (NBS Bio). Purified immunoprecipitated DNA and input DNA were analysed by qPCR using the SsoFast™ EvaGreen® Supermix. See Appendix Table S4 for primers used for amplification.

## Statistics

Graphical data were generated and analysed by GraphPad PRISM software. Statistical analyses were performed using unpaired 2-tailed Student's *t*-test.

## Materials

The following reagents generated for this study are available to request through the MRC-PPU reagents website (https://mrcppurea gents.dundee.ac.uk/): Small molecule inhibitor collection and Brd4 ChIP antibody.

## Data availability

Gene Expression Omnibus GSE60171 [22].

**Expanded View** for this article is available online.

## Acknowledgements

The authors would like to thank Prof. Helen Walden (MRC-PPU, Dundee) and Dr. Tobias Beyer (ETH, Zurich) for critical reading of the manuscript, and Profs. Kate Storey, Doreen Cantrell, Vicky Cowling, and Dr. Kim Dale (School of Life Sciences, University of Dundee) for advice and critical insights. This work was funded by Tenovus Scotland research grant (T11/15). G.M.F. is supported in part by a Medical Research Council New Investigator Award MR/N000609/1, and A.C. is funded in part by a European Research Council Starting Grant ERC-2012-StG-311460.

## Author contributions

GMF and RF-A conceived the study, designed and performed experiments and wrote the paper. LD, JH, ARP and GPS designed and performed experiments and provided conceptual insight. MZ, AL and AC provided reagents and technical expertise.

## Conflict of interest

The authors declare that they have no conflict of interest.

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
