## [Review Process File · EMBO Reports]

Manuscript EMBO-2016-43534

Brd4-Brd2 isoform switching coordinates pluripotent exit and Smad2-dependent lineage specification

Rosalia Fernandez-Alonso, Lindsay Davidson, Jens Hukelmann, Michael Zengerle, Alan R. Prescott, Angus Lamond, Alessio Ciulli, Gopal P. Sapkota, and Greg M. Findlay

Corresponding author: Greg M. Findlay, University of Dundee

Review timeline:

Submission date:	18 October 2016
Editorial Decision:	25 November 2016
Revision received:	27 February 2017
Editorial Decision:	31 March 2017
Revision received:	15 April 2017
Accepted:	24 April 2017

Editor: Achim Breiling

Transaction Report:

1st Editorial Decision

25 November 2016

Thank you for the submission of your research manuscript to EMBO reports. We have now received reports from the three referees that were asked to evaluate your study, which can be found at the end of this email.

As you will see, all three referees acknowledge the potential high interest of the findings and the novelty. However, all three referees have raised a number of concerns and suggestions to improve the manuscript, or to strengthen the data and the conclusions drawn, which need to be addressed during a revision. As the reports are below, I will not detail them here. However, in particular point 1 by referee #1 needs attention (performance and presentation of ChIP data).

Given these constructive comments, we would like to invite you to revise your manuscript with the understanding that all referee concerns must be fully addressed in the revised manuscript and in a complete point-by-point response. Acceptance of your manuscript will depend on a positive outcome of a second round of review. It is EMBO reports policy to allow a single round of revision only and acceptance or rejection of the manuscript will therefore depend on the completeness of your responses included in the next, final version of the manuscript.

REFEREE REPORTS

Referee #1:

Pluripotent stem cells have enormous therapeutic potential with the capacity to differentiate into specialized tissues such as the pancreas, liver, neurons, and cardiac muscle. The molecular mechanisms regulating the exit from pluripotency and towards cellular differentiation remain unknown. The Nodal-Smad signaling pathway plays a crucial role in mesendoderm (ME) development but their mechanisms remain elusive. In pluripotent stem cells autocrine Nodal activated Smad2 signaling reinforcing the expression of pluripotency factors. In differentiating cells, Nodal-Smad2 induces ME-specific genes. In this manuscript, the authors employ a small molecule screen to identify novel regulators of Nodal-Smad2 signaling. Fernandez-Alonso show that the BET bromodomain family proteins are required for Nodal-Smad2 signaling during ME specification. Specifically, they show that Brd4-Brd2 proteins are enriched at Nodal gene regulatory regions. BET protein inhibition reduces phospho-Smad2 expression. Overall, this manuscript is well written and the experiments are well designed to address their hypothesis.

Figure 1 It is difficult to determine which small molecule is being used in C. Please use a different scheme for depicting the CHIR99021 and PD0325901 treatments in the Klf4 and Brachyury graphs. In addition, statistical analysis is necessary for this figure.

The authors show the effectiveness for the inhibitors PD0325901, CHIR99021, and SB505124 in Figure 1D. There is no control for BET inhibitor (JQ1) treatment. Does treatment with JQ1 evict Brd4 by chromatin immunoprecipitation (ChIP) from known pluripotent gene regulatory regions?

It is not clear to me whether the cells in Figure 3B-E are mouse or human. Please specify. Figure 3D Please provide a suitable protein loading control (histone H3, Actin, Lamin, etc) in this rescue experiment.

The authors use BET inhibition to directly modulate hiPSCs to dEN. These terms need defining in the main text..

Figure 4B the lines in the graph are not defined. The images in Figure 4E are not very clear. Please repeat with an increased magnification of a single cell.

Figure 5E Are the western blots in this figure from whole cell extracts? This is not clear. The Brd4 short form immunoblot has no positive control. (The same can be said about Figure 6B anti-Brd4 short) Is there a cell line that can be used to test that this antibody specifically recognized the short form. I'm confused why the immunoblot for the 0 PROTAC treatment in Figure 6 is clearer. In Figure 5F the color scheme for the Brd4 ChIP is difficult to discern. Please use a more contrasting red color in this panel to show 2i versus release conditions.

The Immunoblots in Supplemental Figure 2A and C are not very sharp.

Figure 6. Please show additional ME genes besides Nodal and Brachyury following siRNA.

Referee #2:

This manuscript addresses the molecular mechanisms that coordinate the exit from pluripotency and lineage specification. It specifically focuses on identifying novel regulators of signaling pathways using a small molecule screen. This screen identifies BET Bromodomain proteins as regulators of Nodal/Smad2 signaling in differentiating pluripotent stem cells. The authors propose that the switch between Brd2 and Brd4, two BET protein family proteins, at the Nodal gene regulatory element facilitates the exit from pluripotency and differentiation of cells towards the mesendoderm lineage.

The main criticism of the manuscript is that much of the data is preliminary. The paper reads as an early draft and needs further data to be collected. The paper needs extensive rewriting to make it more readable and bring it up to publication quality. Furthermore, some of the figures are very difficult to follow.

I am not providing an exhaustive list of criticisms. Some examples of the points that could be raised are listed below.

Major Issues:

1. The model the authors present with Brd4 and Brd2 switching is based largely on ChIP data. However the way these experiments have been performed and are presented means the model is questionable. The representation of the ChIP data is very unusual. By comparing everything to IgG control at each locus the reader has no idea of how to compare the data at each position. It would be expected that there should be an IgG control but there should also be a non-binding region (preferably close by but at another locus if necessary). The enrichment should be expressed relative to input. Fig.5 is proposed to show what happens to ChIP signals upon release from 2i. To convince the reader that there is indeed a swap from Brd4 to Brd2 then a time course would be useful. To see a graded swap would be much more convincing. The quantitation at the same position is very different in different panels of fig 5 e. g. panels D and F.
2. How is the proteomic quantitation of the proteins performed? There is no description in the methods. Even if there was a section describing the technique there is then the difficulty of how the authors extrapolate from the data collected in LIF/FCS to the data collected in 2i (fig 5).
3. There are some factual errors in the text e.g. the authors assert that naïve cells undergo multi-lineage differentiation towards epiblast (p4)
4. Most of the error bars in the manuscript are from technical replicates and the data shown are representative experiments. Is there a reason that the data is not collated and the errors shown as standard deviations? At the very least the authors need to note how many times each experiment was performed.
5. Some of the figures are very difficult to interpret. The worst is Fig 1c. It is difficult to see what is meant by this figure. Given it is partly schematic it should be helping the reader understand the data below it but it confuses rather than assists.
6. Given that the work is based on a screen of small molecules, there is too little description of the screen. It would be useful to know more details and to know how the authors take into account toxicity of the compounds.

Some Minor Issues include:

1. Fig 1C. It is unclear, which is PD and which is Chiron as the legend describes them with the same line.
2. Fig 2C. Where are the error bars for the last two samples (RVX-208 and JQ1(-))? Explanation of JQ1(-) condition either in the text or in the figure legend would be useful for the interpretation of the data.
3. Fig 4E. Can authors provide better quality images in order to convince readers that FoxA2 staining is nuclear and lost with the JQ1 treatment?
4. Fig 6D. The x-axis of both graphs are mislabeled (it should be siBrd3 and siBrd4).
5. In a number of cases differences are described as non-significant (e.g. fig 6D). The differences seem large and so the authors need to supply the numbers used to generate these statistics.
6. Could the authors give explanation why the concentration of JQ1 used for rest of the experiments is different from the concentration used in screening experiment?

Referee #3:

This is an interesting paper describing a new role of Brd2-Brd4 in regulating Nodal gene expression. Use of a small molecular inhibitor JQ1, siRNAs and targeted protein degradation of Brd2 all support

the notion. Overall, I think the experiments are done well, but I have the following criticisms.

1. While the authors demonstrate the role of Brd2-4 in the Nodal gene regulation, the mechanism of its action is still unclear. Specifically, what is the underlying molecular mechanism? Do they regulate specific epigenetic modifications or transcriptional machinery of the Nodal gene?
2. Is the Nodal gene the only target gene regulated by Brd2-4? This is important, as the question will address the functional interaction between Brd2-4 and the genome. If Brd2-4 influences the expression of other target genes, how does the current finding fit the data?
3. For the maintenance of pluripotency of human ESCs, Nodal protein is needed, while this is dispensable for mouse ESCs. Will this evidence influence the current interpretation of the data?
4. Figure 6D shows that a 2-fold reduction in siBrd2 is significant, but a 2-fold increase in in siBrd4 is not. I am not convinced that the analysis is done properly. Why are there such large error ranges in Brd4?
5. How does the absence of Brd4 influence the binding of Brd2? Do they work cooperatively or independently?
6. JQ1 treatment affects ES cells differentiation and the expression of definitive endoderm markers in absence (Figure 4). Therefore, I would expect the inhibition of Brd2 to have strong effects on target gene expression. However, the reduction after siRNA is modest. This seems to imply that JQ1 has other effects or that siRNA inhibition is not working well in the current experiment.
7. Figure 1c: The labeling is confusing. The lines representing CHIR99021 and PD0325901 are identical.
8. Figure 4, panel E: FoxA2 staining is very weak, and diffused. I am not so certain whether I am looking at individual colonies or cells.

1st Revision - authors' response

27 February 2017

Referee #1:

Pluripotent stem cells have enormous therapeutic potential with the capacity to differentiate into specialized tissues such as the pancreas, liver, neurons, and cardiac muscle. The molecular mechanisms regulating the exit from pluripotency and towards cellular differentiation remain unknown. The Nodal-Smad signaling pathway plays a crucial role in mesendoderm (ME) development but their mechanisms remain elusive. In pluripotent stem cells autocrine Nodal activated Smad2 signaling reinforcing the expression of pluripotency factors. In differentiating cells, Nodal-Smad2 induces ME-specific genes. In this manuscript, the authors employ a small molecule screen to identify novel regulators of Nodal-Smad2 signaling. Fernandez-Alonso show that the BET bromodomain family proteins are required for Nodal-Smad2 signaling during ME specification. Specifically, they show that Brd4-Brd2 proteins are enriched at Nodal gene regulatory regions. BET protein inhibition reduces phospho-Smad2 expression. Overall, this manuscript is well written and the experiments are well designed to address their hypothesis.

We thank the referee for their positive comments and enthusiasm for our manuscript

Figure 1 It is difficult to determine which small molecule is being used in C. Please use a different scheme for depicting the CHIR99021 and PD0325901 treatments in the Klf4 and Brachyury graphs. In addition, statistical analysis is necessary for this figure.

We agree that original Figure 1C is difficult to follow. Revised Figure 1C is now represented as a bar chart and we provide statistical analysis as requested

The authors show the effectiveness for the inhibitors PD0325901, CHIR99021, and SB505124 in Figure 1D. There is no control for BET inhibitor (JQ1) treatment. Does treatment with JQ1 evict Brd4 by chromatin immunoprecipitation (ChIP) from known pluripotent gene regulatory regions?

We now show in revised Figure EV1C that JQ1 treatment suppresses Nanog, a direct Brd4 target gene. This data confirms that JQ1 effectively evicts Brd4 from known pluripotent gene regulatory regions

It is not clear to me whether the cells in Figure 3B-E are mouse or human. Please specify. Figure 3D

Please provide a suitable protein loading control (histone H3, Actin, Lamin, etc) in this rescue experiment.

We clarify in the text that the cells in Figure 3 are mouse ESCs. In revised Figure 3D we now provide Lamin B1 as an independent protein loading control in addition to Smad2

The authors use BET inhibition to directly modulate hiPSCs to dEN. These terms need defining in the main text.

We thank the reviewer for pointing out this oversight and now define these terms in the revised main text

Figure 4B the lines in the graph are not defined.

We now include these definitions in revised Figure 4B

The images in Figure 4E are not very clear. Please repeat with an increased magnification of a single cell.

In revised Figure 4E we now provide Foxa2 staining at increased resolution to indicate single/individual cells

Figure 5E Are the western blots in this figure from whole cell extracts? This is not clear.

We now confirm in the text that these western blots are from whole cell (i.e. nuclear and cytoplasmic) extracts

The Brd4 short form immunoblot has no positive control. (The same can be said about Figure 6B anti-Brd4 short) Is there a cell line that can be used to test that this antibody specifically recognized the short form. I'm confused why the immunoblot for the 0 PROTAC treatment in Figure 6 is clearer.

For consistency, we present the same exposure for Brd4(S) and Brd4(L) in Figure 5E, although the Brd4(S) signal in this western blot is low as it is relatively poorly expressed compared to Brd4(L). We now provide both short and long exposures in revised Figure 5E to show Brd4(S) more clearly. There is a non-specific band which runs just beneath the specific band for Brd4(S), which we confirm using Brd4-specific siRNA in Figure 6B and PROTAC treatment in Figure 6C. For clarity, we now indicate the specific Brd4(S) band with an asterisk on all immunoblots

In Figure 5F the color scheme for the Brd4 ChIP is difficult to discern. Please use a more contrasting red color in this panel to show 2i versus release conditions.

In the revised Brd4 ChIP figures, we have adjusted the color scheme for clarity

The Immunoblots in Supplemental Figure 2A and C are not very sharp.

We now provide clearer immunoblots in revised Figure EVID and F (original Figures S2A and C)

Figure 6. Please show additional ME genes besides Nodal and Brachyury following siRNA.

We now provide data in revised Figure EV5B for other ME genes Mixl, Lefty1 and Lefty2

Referee #2:

This manuscript addresses the molecular mechanisms that coordinate the exit from pluripotency and lineage specification. It specifically focuses on identifying novel regulators of signaling pathways using a small molecule screen. This screen identifies BET Bromodomain proteins as regulators of Nodal/Smad2 signaling in differentiating pluripotent stem cells. The authors propose that the switch

between Brd2 and Brd4, two BET protein family proteins, at the Nodal gene regulatory element facilitates the exit from pluripotency and differentiation of cells towards the mesendoderm lineage.

The main criticism of the manuscript is that much of the data is preliminary. The paper reads as an early draft and needs further data to be collected. The paper needs extensive rewriting to make it more readable and bring it up to publication quality. Furthermore, some of the figures are very difficult to follow.

We thank the reviewer for their constructive criticisms and identification of areas where the manuscript can be improved. We have now added further data as requested and rewritten the manuscript for clarity

I am not providing an exhaustive list of criticisms. Some examples of the points that could be raised are listed below.

Major Issues:

1. The model the authors present with Brd4 and Brd2 switching is based largely on ChIP data. However the way these experiments have been performed and are presented means the model is questionable. The representation of the ChIP data is very unusual. By comparing everything to IgG control at each locus the reader has no idea of how to compare the data at each position. It would be expected that there should be an IgG control but there should also be a non-binding region (preferably close by but at another locus if necessary). The enrichment should be expressed relative to input.

We now present all ChIP data as enrichment relative to the input, but also include an IgG control for comparison. Furthermore, in revised Figure EV4D we now provide ChIP data for a control genomic region that exhibits low Brd4 binding, as described in Bhagwat et al 2016. This clearly demonstrates that Brd2 and Brd4 are specifically enriched at Nodal promoter elements compared to this control locus

Fig.5 proposes to show what happens to ChIP signals upon release from 2i. To convince the reader that there is indeed a swap from Brd4 to Brd2 then a time course would be useful. To see a graded swap would be much more convincing.

We thank the referee for this suggestion, and agree that a time course examining Brd2 and Brd4 binding to the Nodal promoter during differentiation would be highly informative. We now show in revised Figure 5F that Brd4 recruitment to Nodal is gradually suppressed during mESC differentiation, whilst Brd2 recruitment increases, indicating a graded swap from Brd4 to Brd2 at Nodal gene regulatory regions. We also show that, in 2i mESCs, Brd4 knockdown drives Brd2 recruitment to Nodal promoter elements, further supporting the model that loss of Brd4 binding facilitates Brd2 recruitment

The quantitation at the same position is very different in different panels of fig 5 e. g. panels D and F.

As pointed out above, ChIP quantitation is normalized to IgG control in these original figures, which introduces some variation as the IgG signal is relatively low. We reduce the level of variation by representing ChIP data as percentage of input as requested by the reviewer. Although the percentage enrichment does vary between different experiments due to variations in immunoprecipitation efficiency, data from all experiments presented display the same trend

2. How is the proteomic quantitation of the proteins performed? There is no description in the methods. Even if there was a section describing the technique there is then the difficulty of how the authors extrapolate from the data collected in LIF/FCS to the data collected in 2i (fig 5).

We now provide detailed materials and methods for absolute proteome quantitation of mESCs grown in LIF/FCS. We directly extrapolate copy number data for BET isoforms from LIF/FCS proteomic data by relative quantification of LIF/FCS and 2i/2i release samples from the same immunoblot. In revised Figure 5E, we now provide a complete Brd2-4 immunoblot expression analysis from LIF/FCS and 2i/2i release mESCs, which was used to generate copy number data

3. There are some factual errors in the text e.g. the authors assert that naive cells undergo multi-lineage differentiation towards epiblast (p4)

We thank the reviewer for pointing out this error and correct this statement in the revised manuscript

4. Most of the error bars in the manuscript are from technical replicates and the data shown are representative experiments. Is there a reason that the data is not collated and the errors shown as standard deviations? At the very least the authors need to note how many times each experiment was performed.

We now provide collated data with standard deviation from biological replicates wherever possible. However, in ChIP experiments, variation in immunoprecipitation efficiency between different biological replicates makes it difficult to normalise and therefore collate the data. In these cases we provide a representative experiment and state the number of times each experiment was performed

5. Some of the figures are very difficult to interpret. The worst is Fig 1c. It is difficult to see what is meant by this figure. Given it is partly schematic it should be helping the reader understand the data below it but it confuses rather than assists.

We agree that original Figure 1C is difficult to follow. Revised Figure 1C is now represented as a bar chart and we provide statistical analysis as requested

6. Given that the work is based on a screen of small molecules, there is too little description of the screen. It would be useful to know more details and to know how the authors take into account toxicity of the compounds.

We now provide a more detailed description of our screening methodology in the materials and methods section. We take into account toxicity, and did not analyse Brachyury expression for toxic compounds. We provide a table to summarise this and other primary screening data in the Appendix (Figure S1)

Some Minor Issues include:

1. Fig 1C. It is unclear which is PD and which is Chiron, as the legend describes them with the same line.

We have now redesigned Figure 1C for clarity

2. Fig 2C. Where are the error bars for the last two samples (RVX-208 and JQ1(-))? Explanation of JQ1(-) condition either in the text or in the figure legend would be useful for the interpretation of the data.

We have replaced the error bars in revised Figure 2C, which were deleted in error during formatting. We also explain in the text that JQ1- is an inactive stereoisomer of JQ1

3. Fig 4E. Can authors provide better quality images in order to convince readers that FoxA2 staining is nuclear and lost with the JQ1 treatment?

We now provide Foxa2 staining at increased resolution in revised Figure 4E to confirm that the signal is nuclear and lost upon JQ1 treatment

4. Fig 6D. The x-axis of both graphs are mislabeled (it should be siBrd3 and siBrd4).

We have added the correct labels in revised Figure 6D

5. In a number of cases differences are described as non-significant (e.g. fig 6D). The differences seem large and so the authors need to supply the numbers used to generate these statistics.

We now provide in revised Figure 6D and for all other siRNA experiments the primary data points (technical replicates of multiple experiments) to clearly show the variation in the data. Variation in Nodal and Brachyury expression is relatively high due to variable differentiation efficiency between experiments. Furthermore, Brd4 knockdown actively promotes differentiation, which compounds the variation in Brd4 data. Nevertheless, we show that only Brd2 knockdown produces statistically significant suppression of Nodal/Brachyury expression, although there is a tendency for Brd4 knockdown to increase Nodal/Brachyury expression due to increased differentiation

6. Could the authors give explanation why the concentration of JQ1 used for rest of the experiments is different from the concentration used in screening experiment?

The screening concentration for most compounds in the small molecule library is 1 M. However, JQ1 is a potent and selective BET bromodomain inhibitor with low nanomolar IC50; therefore, when JQ1 was added to the library we selected 100nM as the screening concentration. This was not made clear in the original screening description, but is now shown in the Appendix (Figure S1)

Referee #3:

This is an interesting paper describing a new role of Brd2-Brd4 in regulating Nodal gene expression. Use of a small molecular inhibitor JQ1, siRNAs and targeted protein degradation of Brd2 all support the notion. Overall, I think the experiments are done well, but I have the following criticisms.

We thank the reviewer for their positive comments and enthusiasm for our manuscript

1. While the authors demonstrate the role of Brd2-4 in the Nodal gene regulation, the mechanism of its action is still unclear. Specifically, what is the underlying molecular mechanism? Do they regulate specific epigenetic modifications or transcriptional machinery of the Nodal gene?

We thank the reviewer for these insightful questions and provide new data to show that Brd2-4 binding to the Nodal promoter is required for recruitment of transcriptional machinery and transcriptional elongation. In revised Figure EV4E, we show that nascent transcription from the Nodal gene is inhibited by JQ1 treatment. In revised Figure 5C we show that JQ1 disrupts recruitment of Cdk9, a key factor in transcriptional elongation, to Nodal gene regulatory elements. Finally, we show in revised Figure 5D that JQ1 treatment does not alter H3K4me3 activating epigenetic modifications at the Nodal promoter, further highlighting transcriptional recruitment as a key mechanism of function

2. Is the Nodal gene the only target gene regulated by Brd2-4? This is important, as the question will address the functional interaction between Brd2-4 and the genome. If Brd2-4 influences the expression of other target genes, how does the current finding fit the data?

Brd2-4 regulate expression of a number of genes in ESCs as suggested, including pluripotency factors (reported by di Micco et al, Cell Reports 2014). However, our data pinpoint Nodal as the critical Brd2-4 target gene in the context of Smad2 signalling and during ME differentiation. The key supporting experiments are presented in Figures 3C-E and EV2B, where we show that, even in the presence of JQ1 (i.e. when all Brd2-4 target genes are silenced), recombinant Activin ligand (which activates the same receptors as Nodal) fully rescues Smad2 signalling and ME differentiation. Therefore, other Brd2-4 target genes are not required for Smad2 signalling and ME differentiation. However, we agree that other Brd2-4 target genes are likely to be important at different stages of ESC development e.g. in pluripotency maintenance, proliferation of progenitor cells etc.

3. For the maintenance of pluripotency of human ESCs, Nodal protein is needed, while this is dispensable for mouse ESCs. Will this evidence influence the current interpretation of the data?

**The mechanism we describe is independent of the requirement for Nodal in pluripotency, as both mouse and human cells require Nodal signalling to drive ME differentiation. Interestingly however, we show in Figure EV3A that JQ1 inhibits Nodal-Smad2 signalling, which is required for*

*pluripotency, in undifferentiated hiPSCs (compare lanes 1 & 2). We therefore hypothesise that Nodal is an important BET target gene (amongst other pluripotency genes such as Nanog) in maintaining pluripotency in human cells**

4. Figure 6D shows that a 2-fold reduction in siBrd2 is significant, but a 2-fold increase in in siBrd4 is not. I am not convinced that the analysis is done properly. Why are there such large error ranges in Brd4?

We now provide in revised Figure 6D and for all other siRNA experiments the primary data points (technical replicates of multiple experiments) to clearly show the variation in the data. Variation in Nodal and Brachyury expression is relatively high due to variable differentiation efficiency between experiments. Furthermore, Brd4 knockdown actively promotes differentiation, which compounds the variation in Brd4 data. Nevertheless, we show that only Brd2 knockdown produces statistically significant suppression of Nodal/Brachyury expression, although there is a tendency for Brd4 knockdown to increase Nodal/Brachyury expression due to increased differentiation

5. How does the absence of Brd4 influence the binding of Brd2? Do they work cooperatively or independently?

We provide new data in Figure 6E showing that Brd2 can work independently of Brd4 in recruiting Cdk9 (a key factor in transcriptional elongation) to Nodal gene regulatory elements in 2i release (when Brd4 is removed from Nodal). We also address this question directly by knocking down Brd4 in 2i mESCs, which drives Brd2 recruitment to Nodal promoter elements. These data support the notion that Brd2 and Brd4 function independently during ME differentiation, although we acknowledge that Brd2 and Brd4 may function cooperatively at other promoters and in other cellular contexts

6. JQ1 treatment affects ES cells differentiation and the expression of definitive endoderm markers is absent (Figure 4). Therefore, I would expect the inhibition of Brd2 to have strong effects on target gene expression. However, the reduction after siRNA is modest. This seems to imply that JQ1 has other effects or that siRNA inhibition is not working well in the current experiment.

JQ1 is very selective (no known off-target effects) and completely inhibits all BET bromodomain proteins at 100nM, whilst Brd2 siRNA does not completely ablate Brd2 expression, as shown in revised Figure EV5C (50-75% KD in our experiments). This partial knockdown explains why Nodal and Brachyury gene expression are not completely abolished by Brd2 siRNA. Nevertheless, the magnitude of effect Nodal/Brachyury expression correlates well with Brd2 knockdown

7. Figure 1c: The labeling is confusing. The lines representing CHIR99021 and PD0325901 are identical.

We have redesigned Figure 1C for clarity

8. Figure 4, panel E: FoxA2 staining is very weak, and diffused. I am not so certain whether I am looking at individual colonies or cells.

In revised Figure 4E we now provide Foxa2 staining at increased resolution to indicate single/individual cells

2nd Editorial Decision

31 March 2017

Thank you for the submission of your revised manuscript to EMBO reports. We have now received reports from the three referees that were asked to re-assess your study, which can be found at the end of this email. As you will see, the referees now support the publication of your manuscript. However, referees #2 and #3 have some further suggestions and/or questions that we ask you to address in final revised version of the paper. In addition, I have a couple of further editorial requests:

- The title is currently slightly too long. Please provide a shortened title (less than 100 characters including spaces).

- Please change the references to EMBO Reports style. See:
<http://embor.embopress.org/authorguide#referencesformat>

- Please add a running title and up to five key words to the manuscript text, after the abstract (please change the name of the summary to abstract). Please also remove the bullet points and the two-sentence summary from the main manuscript text. I have saved these separately.

- The table presently in the Appendix is called out in the manuscript text as "Appendix Figure S1". Please change this to "Appendix Table S1", also in the Appendix itself. I suggest to move the 3 tables you now indicated as "Tables and Supplementary Materials" to the Appendix (together with the Table S1). Otherwise, they need to be integrated into the M&M section. Please combine all these tables into one Appendix file (named Appendix) and also add a TOC to the Appendix, including a brief description (title, legend) for each item and page numbers.

- As most of the Western panels are cut and show only a fraction of the original gel, we would prefer to have source data files for these gels (also for the Western panels in the Appendix). Source data is published in a separate source data file online along with the accepted manuscript and will be linked to the relevant figure. Please prepare source data files (containing scans of entire gels or blots) of your experiments including size markers. Please label the scans with figure and panel number, and send one PDF file per figure or per figure panel. We would also appreciate if you could provide higher resolution panels for all the Western blots.

- Finally, please check that all the figures and panels are called out correctly in the text.

REFEREE REPORTS

Referee #1:

The authors have satisfied my concerns and the manuscript is suitable for publication.

Referee #2:

The resubmitted manuscript is greatly improved and the majority of the points raised have been addressed. In particular the ChIP data is presented in a more standard form, the staining images are greatly improved and the confusing figures have been replaced. The experimental details asked for have also been included.

Some remaining points:

- Fig. 1a does not appear to be described in the text
- Fig. 7 is also not referred to.
- The choice of colours for some of the figures is sub-optimal
e.g. Fig 5e (right hand panel), 5g and 6f. The contrast between the shades of grey needs increasing.

Referee #3:

I would like to thank the authors for diligently addressing criticisms. While the authors have addressed most of the criticisms I have raised, I still have the following minor questions. P8 states, "Indeed, chromatin immunoprecipitation (ChIP) for lysine-acetylated Histone H4 enriches for the Nodal HBE (Figure EV4B), confirming that the Nodal promoter is a potential recruitment site for BET family bromodomain proteins." Can you be more specific about the site of histone acetylation? P11. The last sentence of the result section refers to Figure 6G. However, Fig 6G does not exist.

Thank you for inviting us to submit a final revised version of our manuscript to EMBO reports.

We were pleased to hear that the three referees now support publication of our paper. In this final revision, we have now addressed all remaining questions from the referees and your editorial

requests, which I have detailed in the accompanying rebuttal.

 EDITORIAL COMMENTS:

As you will see, the referees now support the publication of your manuscript. However, referees #2 and #3 have some further suggestions and/or questions that we ask you to address in final revised version of the paper.

We have now addressed the referees' remaining questions and thank them for their constructive input

In addition, I have a couple of further editorial requests:

The title is currently slightly too long. Please provide a shortened title (less than 100 characters including spaces).

We now provide a shortened title of less than 100 characters

Please change the references to EMBO Reports style. See:
<http://embor.embopress.org/authorguide#referencesformat>

References have now been reformatted to EMBO Reports style

Please add a running title and up to five key words to the manuscript text, after the abstract (please change the name of the summary to abstract). Please also remove the bullet points and the two-sentence summary from the main manuscript text. I have saved these separately.

We have now changed the summary to abstract, added a running title and five keywords, and removed the bullet points and two sentence summary

The table presently in the Appendix is called out in the manuscript text as "Appendix Figure S1". Please change this to "Appendix Table S1", also in the Appendix itself. I suggest to move the 3 tables you now indicated as "Tables and Supplementary Materials" to the Appendix (together with the Table S1). Otherwise, they need to be integrated into the M&M section. Please combine all these tables into one Appendix file (named Appendix) and also add a TOC to the Appendix, including a brief description (title, legend) for each item and page numbers.

We have now changed Appendix Figure S1 to Appendix Table S1, and removed tables to the Appendix, which now has a TOC and brief description of each item

As most of the Western panels are cut and show only a fraction of the original gel, we would prefer to have source data files for these gels (also for the Western panels in the Appendix). Source data is published in a separate source data file online along with the accepted manuscript and will be linked to the relevant figure. Please prepare source data files (containing scans of entire gels or blots) of your experiments including size markers. Please label the scans with figure and panel number, and send one PDF file per figure or per figure panel. We would also appreciate if you could provide higher resolution panels for all the Western blots.

We now provide full source data consisting of entire scans of all western blots including size markers. We also provide higher resolution versions of the western blot panels

Finally, please check that all the figures and panels are called out correctly in the text.

We confirm that all figure panels are correctly referred to in the text

Referee #1:

The authors have satisfied my concerns and the manuscript is suitable for publication.

We thank Referee 1 for their constructive advice and suggestions

Referee #2:

The resubmitted manuscript is greatly improved and the majority of the points raised have been addressed. In particular the ChIP data is presented in a more standard form, the staining images are greatly improved and the confusing figures have been replaced. The experimental details asked for have also been included.

We thank Referee 2 for their suggestions, which we feel have significantly improved our paper

Some remaining points:

Fig. 1a does not appear to be described in the text

Figure 1A is a model of the experimental system, and is referred to in the introduction

Fig. 7 is also not referred to.

We thank the referee for pointing out this error; in the revised manuscript, Figure 6G is now Figure 7. We have amended this in the text

The choice of colours for some of the figures is sub-optimal, e.g. Fig 5e (right hand panel), 5g and 6f. The contrast between the shades of grey needs increasing.

We have now increased the contrast between shades for clarity

Referee #3:

I would like to thank the authors for diligently addressing criticisms. While the authors have addressed most of the criticisms I have raised, I still have the following minor questions.

We thank Referee 3 for their constructive advice and suggestions

P8 states, "Indeed, chromatin immunoprecipitation (ChIP) for lysine-acetylated Histone H4 enriches for the Nodal HBE (Figure EV4B), confirming that the Nodal promoter is a potential recruitment site for BET family bromodomain proteins." Can you be more specific about the site of histone acetylation?

This antibody recognizes acetylation at multiple lysines within the N-terminal tail of Histone H4 ñ K5, K8, K12 and K16, several of which have been shown to be binding sites for BET bromodomains. We have now specified which Histone H4 acetylation sites are being enriched for in the text and figure legend

P11. The last sentence of the result section refers to Figure 6G. However, Fig 6G does not exist.

We thank the referee for pointing out this error; in the revised manuscript, Figure 6G is now Figure 7. We have amended this in the text

3rd Editorial Decision

24 April 2017

I am very pleased to accept your manuscript for publication in the next available issue of EMBO reports. Thank you for your contribution to our journal.

YOU MUST COMPLETE ALL CELLS WITH A PINK BACKGROUND ↓
 PLEASE NOTE THAT THIS CHECKLIST WILL BE PUBLISHED ALONGSIDE YOUR PAPER

Corresponding Author Name: Greg M. Findlay
Journal Submitted to: EMBO Reports
Manuscript Number: EMBOR-2016-43534V1